# Characteristics and Preparation of Designed Alginate-Based Composite Scaffold Membranes with Decellularized Fibrous Micro-Scaffold Structures from Porcine Skin

**DOI:** 10.3390/polym13203464

**Published:** 2021-10-09

**Authors:** Ching-Cheng Huang

**Affiliations:** 1Department of Biomedical Engineering, Ming-Chuan University, Gui Shan District, Taoyuan 320-33, Taiwan; junas.tw@yahoo.com.tw; 2PARSD Biomedical Material Research Center, Xitun District, Taichung 407-49, Taiwan

**Keywords:** supercritical carbon dioxide, microstructure, decellularization, alginate, composite scaffold membranes

## Abstract

Alginate-based composite scaffold membranes with various ratios of decellularized extracellular matrices could be designed and obtained from porcine skin tissue by using supercritical carbon dioxide fluid technology. Retention of decellularized extracellular matrix (dECM) and scaffold-structure integrity was observed. This work provides a simple and time-saving process for the preparation of biomedical alginate-based composite scaffold membranes with fibrous dECM micro-scaffolds, which were further characterized by Fourier transform infrared spectroscopy (FTIR), thermo-gravimetric analysis (TGA), and scanning electron microscope (SEM). The introduction of fibrous dECM micro-scaffolds enhanced the thermal stability and provided expected effects on the biological properties of the designed composite scaffold membranes in regenerative applications.

## 1. Introduction

Numerous synthetic materials and natural materials have been proposed, modified, and employed for medical applications, such as 3D bioprinting scaffolds for skin and bone tissue reconstruction [1,2,3,4,5,6,7,8,9,10,11,12,13,14,15,16]. Three-dimensional (3D) bioprinting shows potential in tissue engineering and regenerative applications due to its overwhelming advantages over other approaches. In order to promote the functions of bioprinted tissues, the development of novel and versatile bioinks will have crucial implications [1]. Natural materials including agarose, alginate, gellan gum, dextran, hyaluronic acid (HA), silk, fibrin, collagen, decellularized extracellular matrix(dECM), cellulose, gelatin, and chitosan are famous for their excellent biocompatibility and abundance, among which sodium alginate mixed with gelatin has been widely used as a bioink for extrusion-based 3D bioprinting [2]. Despite advances in bioprinting and bio-fabrication during the past decade, fabricating complex and functional tissue constructs that mimic their natural counterparts remains a challenge [3].

Sodium alginate is a naturally occurring biopolymer extracted from different species of marine brown algae [12,13,14,15,16,17]. Sodium alginate, a marine-derived polysaccharide, can offer many advantages over synthetic polymers since they interact under relatively mild temperatures and pH levels [12,13,14,15,16,17]. Alginate shows a wide range of biomedical applications, especially in cell immobilization and tissue regeneration for 3D bioprinting applications, because of its outstanding properties, such as biodegradability and biocompatibility [14]. Alginate has attractive features, including ease of gelation with divalent cations such as calcium ions for enhanced stability of Ca–alginate [12,13,14,15,16,17].

Decellularized extracellular matrix (dECM) scaffolds have abundant levels of collagen, which constitutes the main structural element of the dECM, provides tensile strength, regulates cell adhesion, supports migration, and directs tissue development. Dense connective tissue is an abundant source of dECM scaffolds, which can be prepared and purified by a defatting and decellularizing procedure [4,6,11].

The objectives of the present manuscript are to provide a new design of alginate-based composite scaffold membranes containing micro-scaffolds instead of macroscopic scaffolds and to provide expected effects on the biological properties of the scaffolds in regenerative applications [18,19,20]. Therefore, the designed dECM was prepared by using supercritical carbon dioxide and specific enzymes and were introduced in the composite scaffold membranes to provide a microstructure of micro-scaffolds [4]. The micro-scaffolds could be prepared efficiently by using supercritical fluid treatments instead of macroscopic bioprinting processes. The composite scaffold membranes containing collagen micro-scaffolds must be characterized by Fourier transform infrared spectroscopy (FTIR), thermo-gravimetric analysis (TGA), and scanning electron microscope (SEM) to get the results of identifications, thermal stabilities, and microstructures. A simple method for the evaluation of designed alginate-based composite scaffold membranes with decellularized scaffolds was built up through TGA results (DTG curves) that could easily identify the molecular interactions between alginate, calcium, and the dECM in the resulting composite scaffold membranes. The molecular interactions affected the thermal stability and physicochemical properties of macroscopic scaffolds. Furthermore, the introduction of micro-scaffolds into the composite scaffold membranes will be important for 3D bioprinting or regenerative applications.

## 2. Materials and Methods

### 2.1. Materials

The chemicals used in this work include the enzyme papain (Sigma-Aldrich Company, St. Louis, MO, USA), sodium alginate (Sigma-Aldrich Company), calcium chloride (Fluka Chemie GmbH, Buchs, Switzerland), Triton X-100 (*t*-octylphenoxypolyethoxyethanol) (Sigma-Aldrich Company), and NaOH (Sigma-Aldrich Company).

### 2.2. Preparation of a Decellularized Extracellular Matrix Scaffold

The supercritical fluid of carbon dioxide (ScCO_2_) was employed for the preparation of newly designed decellularized extracellular matrix scaffolds in this study. The ScCO_2_ was employed before the enzyme treatments to remove most fatty acids and tissues [4]. The skin was placed in a tissue holder that was then placed into a ScCO_2_ vessel system. The ScCO_2_ system was then operated at 350 bar and 35 °C for 2 h.

The steady thickness of about 0.5 mm of thinly sliced tissue samples was obtained from the porcine dermal layer by using a designed tissue-cutting machine (Taiwan PARSD Pharm. Tech. Consulting Ltd., Co., Taichung, Taiwan) and Kuin Biotech. Ltd., Co., Xinbei, Taiwan). Residual fat tissues were cleaned from the skin, and the skin was washed with phosphate-buffered saline (PBS). The ScCO_2_ extraction was performed for complete decellularization. Samples were soaked in 25% NaOH_(aq)_ for 2 h with a magnet mixer and then washed in 0.5 U/mL papain(aq) at 25 °C for 2 h [6]. The resulting samples were washed with double-distilled water under ultrasonic waves to remove residual fat and organic matter. The resulting sample was frozen for 6 h and then lyophilized (EYELA, FD-5N) overnight with the use of a freeze dryer at 0.1–0.2 torr at a freeze-drying temperature of −45 °C. A designed decellularized extracellular matrix scaffold, dECM, was then be obtained.

### 2.3. Preparation of Decellularized Extracellular Matrix/Alginate Composite Scaffold Membranes

In this study, composite scaffold membranes with dECM scaffolds were prepared based on the various weight ratios of alginate and dECM (alginate/dECM: 100/0, 95/5, 90/10, 85/15, 80/20). Briefly, the desired amount of dECM powder was first dispersed completely in 40 mL of double-distilled water with a homogenizer at 26,000 rpm(revolutions per minute) for 3 min. Then, an alginate aqueous solution was homogenized thoroughly with the dispersed dECM solution at 26,000 rpm for 3 min. The alginate/dECM solutions were molded and frozen for 6 h and then lyophilized (EYELA, FD-5N) overnight. New alginate/dECM composite scaffold membranes were obtained, such as AdE1N, AdE2N, AdE3N, and AdE4N (Table 1).

### 2.4. Preparation of Cross-Linked Decellularized Extracellular Matrix/Alginate Composite Scaffold Membranes

The decellularized extracellular matrix/alginate composite scaffold membranes were soaked in CaCl_2_ aqueous solutions with various concentrations for different cross-linking reactions with a magnet mixer. The cross-linked decellularized extracellular matrix/alginate composite scaffold membranes were then molded, frozen, and dried by the same procedure described above. Designed decellularized extracellular matrix/alginate composite scaffold membranes were obtained (Table 1).

### 2.5. Measurements

Fourier Transform Infrared (FTIR) spectra were recorded with a spectrometer (Nicolet IS10, Thermo Fisher Scientific, USA) using KBr discs and collecting data from 400–4000 cm^−1^. Thermal analysis was performed by Thermogravimetry Analysis (TGA) using a thermoanalyzer (7300TG/DTA, Seiko, Japan). All measurements employed a linear heating rate of 10 °C·min^−1^, with nitrogen as carrier gas and a platinum empty pan as a reference material. Diameters of micro-scaffolds were studied by scanning electron microscopy (SEM) (S3400N, Hitachi, Japan).

## 3. Results

### 3.1. Fourier Transform Infrared Spectroscopy Analysis of Alginate/dECM Composite Scaffold Membranes

From the FTIR analysis of the original porcine skin (Figure 1A), absorption bands at 1452, 1400, 1337, 1240, 1203, and 1080 cm^−1^ were attributed to amide III, containing δ(CH_2_), δ(CH_3_), ν(C–N), and δ(N–H) absorptions of collagen in the original porcine skin. Amide I and amide II absorptions were found at 1632 and 1551 cm^−1^, respectively. The absorption band at 3301 cm^−1^ δ(C–H) was attributed to the fatty acid of the original porcine skin. The absorption band at 1744 cm^−1^ δ(C=O) was attributed to the fatty acid. The absorption bands of fatty acids were not observed in Figure 1B, which demonstrates the effectiveness of the supercritical carbon dioxide treatment.

Figure 1C shows the typical absorption bands of sodium alginate, mainly the O–H stretching at 3424 cm^−1^, pyranoid ring (six-membered ring) C–H stretching at 2903 and 2932 cm^−1^, COO asymmetric stretching at 1595 cm^−1^, COO symmetric stretching at 1408 cm^−1^, C–O stretching at 1338 and 1298 cm^−1^, and C–O–C stretching at 1094 cm^−1^. Figure 1D still shows a remarkable absorption band at 2903 cm^−1^ on the spectra of ALG/dECM membranes, which indicates that the formation of the egg-box model due to cross-linking with calcium ions did not occur. If the formation of the egg-box model had occurred, the stretching vibration of C–H in the six-membered ring of the calcium alginate molecule would be limited, and the corresponding absorption band could not be observed on the spectrum. In the sodium alginate molecule, the stretching vibration absorption band of –COO– and C–O is very weak. However, in the calcium alginate molecule, the –C–O–O–Ca–O–CO– structure makes the C–O stretching vibration absorption increase and creates an obvious absorption band at 1024 cm^−1^, as shown in Figure 1C,D, which indicates the formation of the –CO–O–Ca–O–CO– structure in ALG/dECM membranes.

The FTIR spectroscopy analysis was carried out to confirm the incorporation of dECM in ALG/dECM composite scaffold membranes. The spectrum of the ALG/dECM composite scaffold membrane (AdE4H) (Figure 1D), besides retaining the above-mentioned bands of pure sodium alginate (AdE0N) (Figure 1C), showed a stronger absorption band at 1595 cm^−1^ (the carbonyl (C=O) bond) and two remarkable shoulders at 1632 cm^−1^ and 1537 cm^−1^, which were characteristic absorption bands of carbonyl groups of amide and dECM molecules, which confirmed the formation of ALG/dECM composite scaffold membrane effectively. For the ALG/dECM composite scaffold membrane, the main absorption bands were also observed at around 1632 cm^−1^ (amide I, C–O, and C–N stretching), 1537 cm^−1^ (amide II), and 1242 cm^−1^ (amide III). There was higher absorption from 3600 cm^−1^ to 3200 cm^−1^ that appeared on the spectrum of the ALG/dECM scaffolds. This suggests an increase in hydrogen bonds resulting from the interaction between the dECM molecule and alginate (ALG) molecule. The results of FTIR indicate the presence of dECM in the hybrid scaffold as wells as the interaction between them.

### 3.2. The Microstructure of Resulting Alginate/dECM Composite Scaffold Membranes

The microstructures of resulting membranes with dECM scaffolds were characterized by a scanning electron microscope (SEM). Scanning electron micrographs of original porcine skin, pretreated porcine skin by supercritical carbon dioxide, and dECM after treatment with supercritical carbon dioxide were showed in Figure 2A–C, respectively. Most of the residual fat tissues were removed to obtain a micro-scaffold structure. After the entire decellularization procedure, the clear micro-scaffold with a fibrous structure was observed directly in the dECM sample derived from porcine skin in the micrometer scale (Figure 2C). The diameter of the fibrous structure was found to range from 8 to 25 µm (Figure 2C). The different micro-scaffolds shape relatively narrow boundaries while sheet structures of ALG samples were observed in SEM, as shown in Figure 2D. The diameter of the narrow boundaries was found to range from 1 to 3 µm. Macroscopic images of the composite scaffold membranes AdE4N and AdE4L were obtained in Figure 3. The micro-scaffold structures of the resulting alginate/dECM composite scaffold membranes were observed clearly whether the cross-linking reaction of CaCl_2_ was carried out or not. Furthermore, scanning electron micrographs of resulting new decellularized composite scaffold membranes are shown in Figure 4A–D. The remarkable micro-scaffolds with fibrous structures were observed in the composite scaffold membranes with various ratios of dECM and ALG. The merged structures of AdE1H and AdE2H with low dECM/ALG ratios were observed in Figure 4A,B. The merged structures combined the sheet structure of ALG molecules with the fibrous structure of dECM molecules. With the increasing introduction ratio of dECM to ALG, the new combined micro-scaffold shapes were observed with smooth fibrous structures, as shown in Figure 4C,D. The diameter of the smooth fibrous structures was found to range from 5 to 30 µm, which was similar to the micro-scaffold shape of dECM (Figure 2C).

### 3.3. Thermal Stability of Resulting Alginate/dECM Composite Scaffold Memebranes

The thermal stability of resulting alginate/dECM composite scaffold membranes could be characterized by TGA. The peak temperatures of the derivative thermogravimetry (DTG) curves are summarized in Table 2. Due to the addition of each of the three components (Alginate + dECM + Calcium), there were three distinct peaks occurring and labelled as a, b, and c (Table 2 and Figure 5). The maximum pyrolysis temperature (T_d__max_) of the original alginate material is lower than 250 °C. To enhance the thermal stability of the composite scaffold membranes, dECM was introduced. The T_d__max_ of dECM is higher than 300 °C. The resulting composite scaffold membranes with dECM molecules would be designed as a new heat-resistant biomaterial. TGA analyses of ALG membranes with and without CaCl_2_ (AdE0N, AdE0H, AdE0L) were determined. The main loss was presented in three different temperature ranges, given by I (<200 °C), II (200–370 °C) and III (370–500 °C). Initial weight loss up to 200 °C was found to be 15%, 18%, and 20% for AdE0N, AdE0H, and AdE0L, respectively, due to the elimination of absorbed and bounded water molecules within the membrane. This increase in weight loss may be due to more adsorption of water molecules present along with Ca^2+^ molecules while cross-linking with CaCl_2_ aqueous solution. However, more interestingly, we observed that for the dECM loaded ALG/dECM membrane samples, the weight loss was less than 10%. Furthermore, the second stage (stage II) of weight loss was observed from 200 to 370 °C and corresponds to thermal degradation of ECM molecules due to the breakage of the protein chains. The relatively high T_dmax_(b) values of ALG/dECM were observed over 350 °C compared to 330 °C for the dECM molecule. Additionally, the relatively high T_dmax_(a) values of non-cross-linked ALG/dECM composite scaffold membranes (such as AdE1N, AdE2N, AdE3N, and AdE4N) were observed to range from 260 to 300 °C compared to 245 °C for the non-cross-linked ALG molecule (Table 2).

When CaCl_2_ (5 wt%) was added into the low dECM-loaded ALG/dECM membrane samples, the relatively high T_dmax_(a) of the cross-linked ALG molecule in AdE1H was observed at 298 °C compared to 245 °C for the non-cross-linked ALG molecule in AdE1N and 270 °C for the slight cross-linked ALG molecule in AdE1L (Figure 5A–C). This was due to the cross-linking reaction among CaCl_2_ and multiple ALG molecules. A new T_dmax_(c) was observed at 390 °C in Figure 5C, which might be due to the cross-linking reaction among CaCl_2_ and multiple dECM molecules. Similarly, a relatively high T_dmax_(a) was observed in a high dECM-loaded AdE4H with a high addition of CaCl_2_ (5 wt%) compared to AdE4N (260 °C) and AdE4L (285 °C) (Figure 5D–F and Table 2). Particularly, a high T_dmax_(c) at 420 °C was observed on the DTG curve of AdE4H (Figure 5F), which might be due to the composite associations and cross-linking reaction among CaCl_2_, multiple ALG molecules, and multiple dECM molecules. New mixed cross-linked network microstructures of ALG molecules and dECM molecules might be formed.

When a small amount of dECM was introduced into the ALG/dECM composite scaffold membrane without CaCl_2_, a weak ionic association between –COOH groups of ALG molecules and –NH_2_ groups of dECM molecules was formed. Therefore, it was difficult to build up the cross-linking structure. With an increasing addition of dECM to ALG/dECM composite scaffold membranes, the ordinary ionic association between –COOH groups of ALG molecules and –NH_2_ groups of dECM molecules was employed to build a weak ionic cross-linking microstructure. When a large amount of dECM was introduced into the ALG/dECM composite scaffold membrane without CaCl_2_, a strong ionic association between –COOH groups of ALG molecules and –NH_2_ groups of dECM molecules was employed to build up the strong ionic cross-linking microstructure. The addition of CaCl_2_ induced various ionic associations among –COOH groups of ALG molecules, –COOH groups of dECM molecules, and Ca^2+^ ions. The remarkably high T_dmax_(c) was observed to be over 400 °C (Table 2). It is important to mention that most of the weight loss occurred at temperatures above human body temperature, which indicates that all these structures will be very stable under bodily conditions. This certainly showed that the introduction and cross-linking of dECM and alginate increased resistance to pyrolysis and provided stability to the composite scaffold membranes that could be considered to be a suitable potential material for soft tissue implants and bioprinting applications [21].

## 4. Conclusions

In this study, new composite scaffold membranes with collagen scaffolds were successfully obtained from alginate and dECM that was prepared from porcine skin using supercritical carbon dioxide fluid technology. The retained extracellular matrix and integrity scaffold structures were observed. This work provides a simple and time-saving method to process decellularized tissue. The network–scaffold microstructures were observed in new composite scaffold membranes with collagen scaffolds. Composite scaffold membranes with high thermal stability were obtained, which indicates that these membranes will be very stable under bodily conditions. The resulting composite scaffold membranes with fibrous micro-scaffold structures could be considered to be a potential material for bioprinting applications.

## Figures and Tables

**Figure 1 polymers-13-03464-f001:**
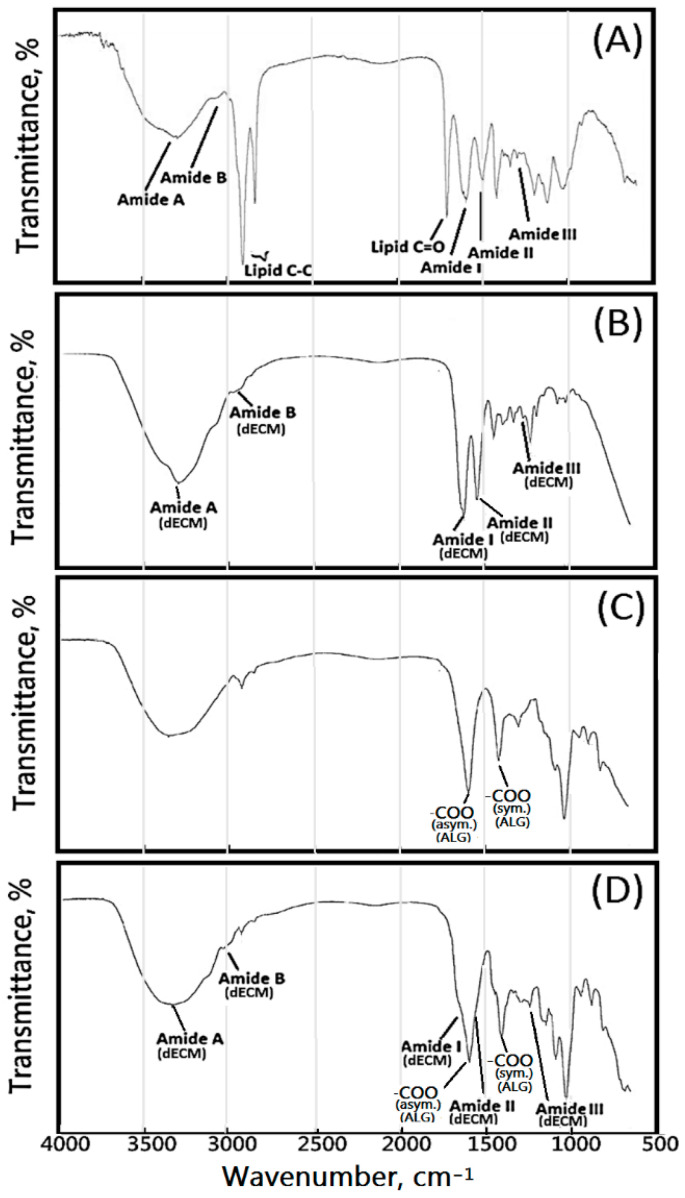
FTIR spectrum of the samples: (**A**) original porcine skin, (**B**) dECM, (**C**) AdE0N, and (**D**) cross-linked ALG/dECM composite scaffold membrane (AdE4H).

**Figure 2 polymers-13-03464-f002:**
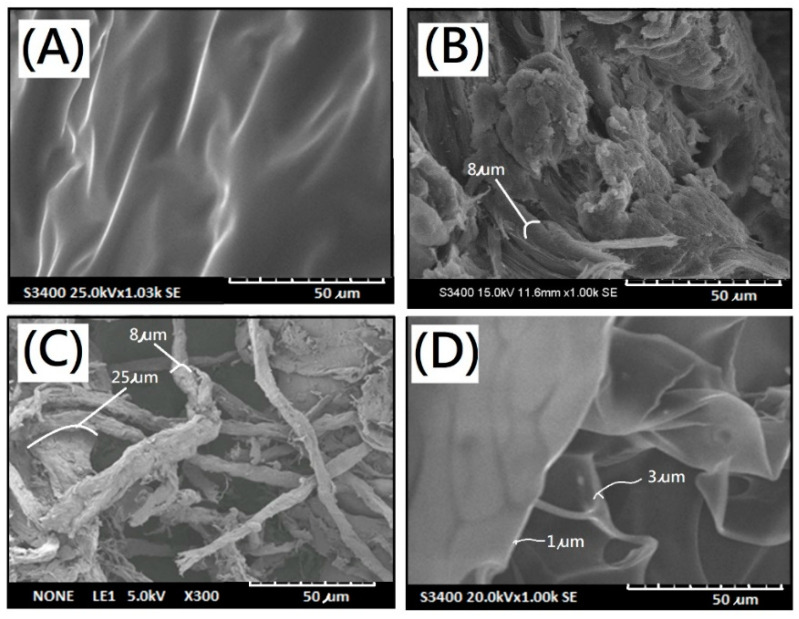
Scanning electron micrographs of samples of (**A**) original porcine skin, (**B**) pretreated porcine skin by ScCO_2_, (**C**) dECM powder, and (**D**) AdE0N membrane.

**Figure 3 polymers-13-03464-f003:**
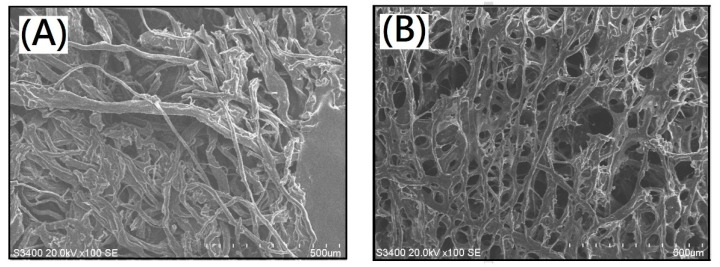
Macroscopic images of the composite scaffold membranes (**A**) AdE4N and (**B**) AdE4L.

**Figure 4 polymers-13-03464-f004:**
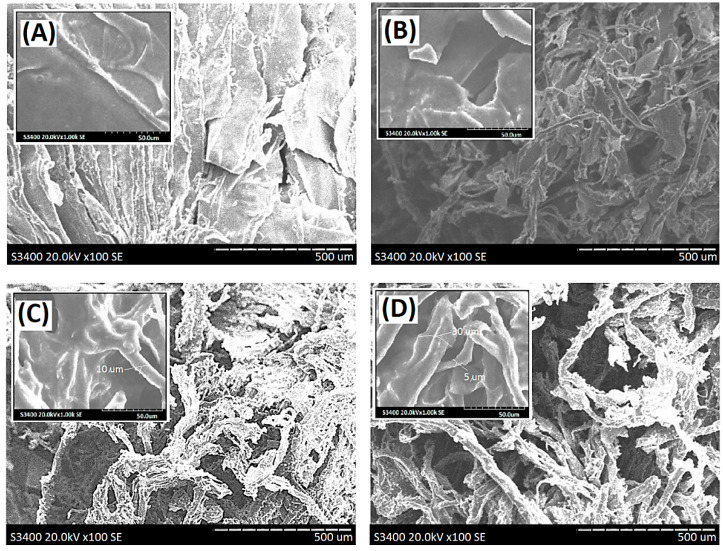
Scanning electron micrographs of the ALG/dECM composite scaffold membrane samples (**A**) AdE1H, (**B**) AdE2H, (**C**) AdE3H, and (**D**) AdE4H.

**Figure 5 polymers-13-03464-f005:**
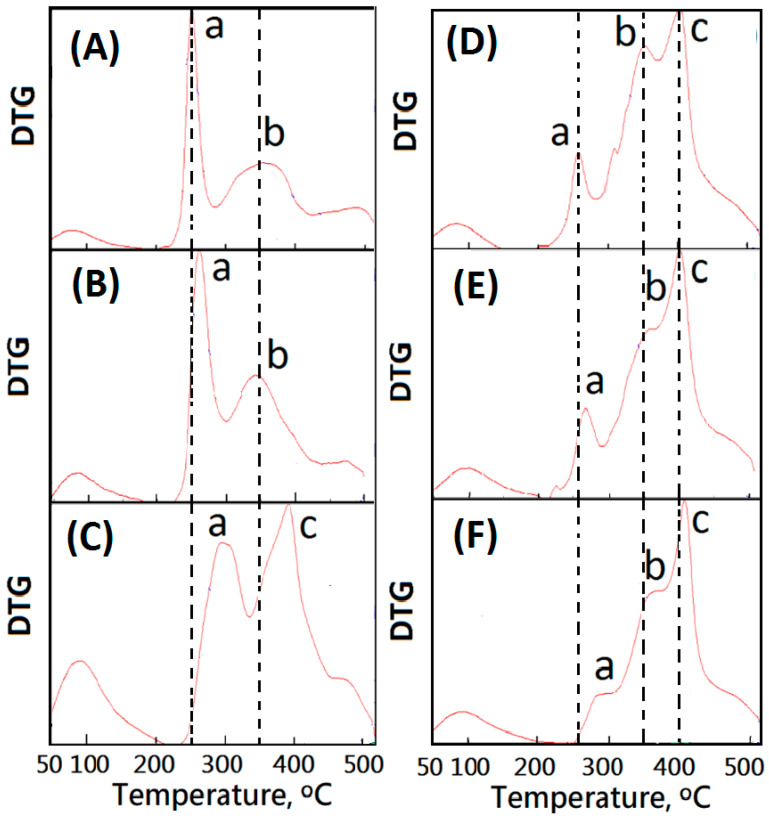
DTG curves of the composite scaffold membranes cross-linked with various concentrations of CaCl_2(aq)_: (**A**) AdE1N(0 wt%), (**B**)AdE1L(1 wt%), (**C**)AdE1H(5 wt%), (**D**) AdE4N(0 wt%), (**E**) AdE4L(1 wt%), and (**F**) AdE4H(5 wt%).

**Table 1 polymers-13-03464-t001:** Preparation of new ALG/dECM composite scaffold membranes.

Sample No.	ALG/dECM ^a^ (*w*/*w*)	CaCl_2_
AdE0N ^a^	100/0	-
AdE1N ^a^	95/5	-
AdE2N ^a^	90/10	-
AdE3N ^a^	85/15	-
AdE4N ^a^	80/20	-
AdE0L ^a,b^	100/0	1 wt%
AdE1L ^a,b^	95/5	1 wt%
AdE2L ^a,b^	90/10	1 wt%
AdE3L ^a,b^	85/15	1 wt%
AdE4L ^a,b^	80/20	1 wt%
AdE0H ^a,c^	100/0	5 wt%
AdE1H ^a,c^	95/5	5 wt%
AdE2H ^a,c^	90/10	5 wt%
AdE3H ^a,c^	85/15	5 wt%
AdE4H ^a,c^	80/20	5 wt%

^a^ ALG(A): alginate and dECM(dE): porcine skins via ScCO_2_ and papain treatments.; ^b^ Relatively low concentration of CaCl_2 (aq)_; ^c^ Relatively high concentration of CaCl_2 (aq)_.

**Table 2 polymers-13-03464-t002:** TGA analysis of ALG/dECM composite scaffold membranes.

Sample No.	a	b	c
AdE0N ^(a)^	245 °C	─ ^(d)^	─ ^(d)^
AdE1N ^(a)^	260 °C (32 wt%)	360 °C (11 wt%)	─ ^(d)^
AdE2N ^(a)^	260 °C (28 wt%)	360 °C (15 wt%)	─ ^(d)^
AdE3N ^(a)^	260 °C (10 wt%)	360 °C (26 wt%)	─ ^(d)^
AdE4N ^(a)^	260 °C (8 wt%)	360 °C (27 wt%)	─ ^(d)^
dE ^(a)^	─ ^(d)^	330	─ ^(d)^
AdE0L ^(a,b)^	260 °C	─ ^(d)^	─ ^(d)^
AdE1L ^(a,b)^	265 °C (20 wt%)	350 °C (35 wt%)	─ ^(d)^
AdE2L ^(a,b)^	268 °C (15 wt%)	360 °C (23 wt%)	410 °C (17 wt%)
AdE3L ^(a,b)^	280 °C (12 wt%)	360 °C (24 wt%)	410 °C (19 wt%)
AdE4L ^(a,b)^	285 °C (5 wt%)	370 °C (25 wt%)	410 °C (25 wt%)
AdE0H ^(a,c)^	280 °C	─ ^(d)^	─ ^(d)^
AdE1H ^(a,c)^	298 °C (15 wt%)	─ ^(d)^	390 °C (20 wt%)
AdE2H ^(a,c)^	298 °C (9 wt%)	360 °C (18 wt%)	410 °C (22 wt%)
AdE3H ^(a,c)^	298 °C (6 wt%)	360 °C (18 wt%)	410 °C (27 wt%)
AdE4H ^(a,c)^	300 °C (5 wt%)	385 °C (18 wt%)	420 °C (30 wt%)

^(a)^ ALG(A): alginate and dECM(dE): Porcine skins via ScCO_2_ and papain treatments; ^(b)^ 1.0 wt% CaCl_2 (aq)_; ^(c)^ 5.0 wt% CaCl_2 (aq)_; ^(d)^ No peak in DTG curve.

## Data Availability

Data available on request due to restrictions eg privacy or ethical.

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
