# Peer review of "Characteristics and Preparation of Designed Alginate-Based Composite Scaffold Membranes with Decellularized Fibrous Micro-Scaffold Structures from Porcine Skin"

_polymers, 2021, doi:10.3390/polym13203464_

Round 1
Reviewer 1 Report
In this study, novel composite membranes with alginate and decellularized scaffolds were obtained from porcine skin tissue. Supercritical carbon dioxide extraction process was employed to prepare decellularized extracellular matrix scaffolds. The composite membranes were characterized by FTIR, SEM, and TGA. The composite membranes exhibited favorable microstructure and high thermal stability. The manuscript is both interesting and relevant. In my opinion, it can be published after major revision:
- The Abstract lacks of specific results. It is expected that the abstract addresses specific findings and quantitative results.
- Also in the abstract, the use of alginate in the composite membranes is not described.
- In section 2.2 the procedure for the supercritical extraction process must be explained in more detail.
- There is a typo in the title of Table 1. There is also a typo in the header of section 3.1. Also there is a typo in page 7, line 163.
- Section 3.2. Please explain how the average diameter of microscaffolds was obtained.
- Also in section 3.2, page 8 line 174-175 it is stated that: “The new combined micro-scaffold shapes could be proposed in the schematic diagrams as shown in Figure 4 and Figure 5.” But Figure 4 and Figure 5 does not have information of schematic diagrams of the new combined scaffold shapes. They have information of TGA analysis.
- In secion 3.3 Line 248 it is stated that “Tdmax(c) value of ALG/dECM could not be found in Figure 6(c).” I guess it should be Figure 6(a)
- Also in section 3.3 lines 214, 232, 235, 242, and 283, the acronym ECM is not defined
- In the section “Conclusions” There is no evidence shown in the manuscript that “the resulting composite membranes with scaffold microstructures could be a potential application for gene therapy.”
Author Response
- The specific findings and quantitative results had been added in the revised manuscript as the reviewer I’s suggestion.
2.The use of alginate in the composite scaffold membranes had been described clearly in the Abstract of revised manuscript as the reviewer I’s suggestion.
- The procedure for the supercritical extraction process had been explained in more detail in section 2.2 of revised manuscript as the reviewer I’s suggestion.
- As reviewer I’s suggestion, the typo in Table 1, the typo in the header of section 3.1, and the typo in page 7, line 163 had been corrected. Also, the typo had been checked and corrected in the revised manuscript.
5.The description of “averaged diameter “is not suitable and modified(line 193 and line 207 in the revised manuscript). As the reviewer I’s suggestion, the determination of diameter of micro-scaffolds had been showed in the modified Figures 2(B)~2(D) of the revised manuscript.
- The statement “The new combined micro-scaffold shapes could be proposed in the schematic diagrams as shown in Figure 4 and Figure 5.”was deleted in the revised manuscript because Figure 4 and Figure 5 did not have information of schematic diagrams of the new combined scaffold shapes as reviewer I’s suggestion.
- The statement of“Tdmax(c) value of ALG/dECM could not be found in Figure 6(c).”was corrected to ” Tdmax(c) value of ALG/dECM could not be found in Figure 7(a).”in the revised manuscript (line 297) as reviewer I’s suggestion.
8 As the reviewer I’s suggestion, the acronym ECM in in section 3.3 lines 214, 232, 235, 242, and 283 in the original manuscript were corrected (the section 3.3 in the revised manuscript).
9 The statement “the resulting composite scaffold membranes with microscaffold structures could be a potential application for gene therapy.” was modified to ” . The resulting composite scaffold membranes with micro-scaffold structures could be considered as a potential material for biomedical applications.” In the revised manuscript as the reviewer I’s suggestion. (lines 341-343 in the revised manuscript)
Reviewer 2 Report
Good article.
Abstract should be improved, incorporating the main results obtained in the study.
- – “employed” instead” employed” in line 37
- - pH instead Ph in line 42.
Author Response
- As the reviewer II’s suggestion, “employed” instead” employed” in line 37 of original manuscript.(line 45 in the revised manuscript)
- As the reviewer II’s suggestion, “Ph” instead “Ph” in line 42 of original manuscript. (line 50 in the revised manuscript)
Reviewer 3 Report
The author presents a study on the preparation of an alginate hydrogel combined with a decellularized skin sample (dECM). The title implies the generation of a bioink, however, no bioink is shown, or any other 3D structure/scaffold that could be potentially used. In general, the study is weak and it is difficult to understand what the author really wants to show. Furthermore, the proposed models of interactions between the dECM and alginate together with calcium ions is more like a personal idea and does not rely on enough experimental evidence. Solely doing TGA, basic FTIR is not enough to study the interaction. At least zeta-potential measurements, viscosity and mechanical properties of the final scaffold should have been performed.
Also, the use of supercritical CO2 (ScCO2), as highlighted in the title, sounds promising, but it is not really shown how its use helps to be beneficial to obtain dECM.
In general, the experimental section is very weak, especially the section about the supercritical CO2 is just two sentences, and it is not clear how the author exactly used this method. Also the cited reference is not related to ScCO2 at all. Some methods are completely missing in the M&M section, such as FTIR and SEM.
In the results section, the SEM pictures do not show much. At least a macroscopic picture of a scaffold should be shown. The description is in general bad, sometimes describing wrong and unclear things (e.g. "The averaged diameter of microscafford was found in a range of 8~25 μm"). The diameter of what? Furthermore, the thickness of boundaries (e.g. line 166) within a dried alginate scaffold does not make sense, since a dried scaffold does not represent the real case, when a scaffold is hydrated. The main part of the study is based on the TGA analysis of the scaffolds. However, it is not clear why it is so important to study the temperature dependence, only to interpret molecular interactions between alginate, calcium and the dECM. Other methods would be more interesting for 3D printing, such as viscosity and mechanical stability.
With all the above mentioned issues, I suggest to reject this manuscript.
Author Response
- As the reviewer III’s comments about “The title implies the generation of a bioink, however, no bioink is shown, or any other 3D structure/scaffold that could be potentially used”. The study would like to build up an evaluated method for design of bioinks by using TGA, FTIR, and SEM. The interactions of different molecules in the printed composite materials could be effective identification. The title had been modified to match the aim and results of the manuscript. The word “bioink” had been deleted in the revised manuscript.
- As the reviewer III’s comments, the study would like to build up an additional evaluated method for design of bioinks by using TGA, FTIR, and SEM except of zeta-potential measurements, viscosity and mechanical properties. The study focuses on the characterization of bioprinted composite scaffold membranes instead of bioinks. The characterized methods of zeta-potential measurements, viscosity and mechanical properties are important for formulations of bioink solutions or hydrogel as the reviewer’s comments. The word “bioink” had been deleted in the title of revised manuscript to avoid confusion.
- In this study, the supercritical CO2 (ScCO2) was employed to obtain micro-scaffolds in the target composite scaffold membranes as highlighted in the title. As the reviewer III’s comments, the evident for help of ScCO2 had been added as shown in Figure 2 (B) of revised manuscript. The ScCO2 could remove most of the residue fat molecules to get micro-scaffolds.
- As the reviewer III’s suggestion, the experimental section had been modified and the reference 4 related to ScCO2 had been cited in the revised manuscript. The methods such as FTIR and SEM had been added in the M&M section of manuscript.
- As the reviewer III’s suggestion, the macroscopic picture had been added in the Figure 3 of revised manuscript.The SEM were important for the observation of micro-scaffold in this study.
6.As the reviewer III’s suggestion, the descriptions had been modified to avoid confusion such as "The averaged diameter of microscafford was found in a range of 8~25 μm"). The study would like to provide a composite scaffold membrane containing micro-scaffolds instead of macroscoipic scaffolds. Therefore, the designed dECM had been prepared and introduced in the composite scaffold membrane to provide a microstructure of micro-scaffolds. In this study, the micro-scaffolds could be prepared efficiently by using supercritical fluid treatments instead of macroscoipic bioprinting scaffolds. As we know, the introductuion of micro- scaffolds in the bioprinted composted materials are important for regenerative applications. The composite scaffold membrane would be prepared by lyophilisation as experimental statements. The thickness of boundaries of micro-scaffolds within a dried alginate-based composite foam membrane could be obtained. The thickness of boundaries of micro-scaffolds is quite different from those of dried macroscoipic alginate scaffold (hydrated state) as reviewer III’s comments. The statements were not clear as reviewer III’s comments. The statements about “macroscoipic alginate scaffold” and “micro-scaffold” had been modified and described clearly to avoid confusion.
7.As the reviewer III’s suggestion, the descriptions about thermal analysis had been modified and stated clearly to avoid confusion. It is important to build up a simple method for evaluation of designed bioprintable materials and bioprinted materials derived from bioinks. In this study, the thermal analysis was employed and clear and easy identification was obtained through TGA results(DTG curves). The results could easily identify the molecular interactions between alginate, calcium and the dECM in the bioprinted membrane, which affect the stability and physicochemical properties( such as zeta-potential, viscosity and mechanical stability) of the bioprinted macroscopic scaffold and bioinks.
Round 2
Reviewer 1 Report
The authors have addressed all the issues raised. In my opinion, the paper is recommended for publication
Author Response
As Reviewer' comments, the manuscript had been checked and modified.
Reviewer 3 Report
Answers:
I thank the author for addressing my comments. However, I am still skeptical about the quality of this study. The introdcutino is really messy, and the author is not coming to the main point of the study. In almost every paragraph in the introduction, the aim of the study is described. This should be done in the last paraghraph, clearly describing the aim of the study. At the moment, there are too many repetitions and it is difficult to read.
Please also see my answers to the author's statements followed by additional comments:
- As the reviewer III’s comments about “The title implies the generation of a bioink, however, no bioink is shown, or any other 3D structure/scaffold that could be potentially used”. The study would like to build up an evaluated method for design of bioinks by using TGA, FTIR, and SEM. The interactions of different molecules in the printed composite materials could be effective identification. The title had been modified to match the aim and results of the manuscript. The word “bioink” had been deleted in the revised manuscript.
Still, the title contains "bioprintable", but it has not been shown in the paper, that the system is bioprintable. I would at least suggest to include something like "for bioprinting applications" or similar.
If the presented system of alginate and dECM should be bioprintable, the study should contain some information on viscosity, at least. I know, that the viscosity of alginate can be tuned by adjusting the concentration or by the addition of calcium, but at the present status, it seems to be more like a hydrogel. But in the presented manuscript, only films of alginate and dECM is presented.
- As the reviewer III’s comments, the study would like to build up an additional evaluated method for design of bioinks by using TGA, FTIR, and SEM except of zeta-potential measurements, viscosity and mechanical properties. The study focuses on the characterization of bioprinted composite scaffold membranes instead of bioinks. The characterized methods of zeta-potential measurements, viscosity and mechanical properties are important for formulations of bioink solutions or hydrogel as the reviewer’s comments. The word “bioink” had been deleted in the title of revised manuscript to avoid confusion.
Are the composite membranes bioprinted? If so, please include the methods, e.g. which bioprinter has been used? In the methods section, the composite solutions were casted in a mold an freeze dried. I would not call this bioprinting.
- In this study, the supercritical CO2 (ScCO2) was employed to obtain micro-scaffolds in the target composite scaffold membranes as highlighted in the title. As the reviewer III’s comments, the evident for help of ScCO2 had been added as shown in Figure 2 (B) of revised manuscript. The ScCO2 could remove most of the residue fat molecules to get micro-scaffolds.
I agree with the author that the use of ScCO2 seems to be a good method to remove cellular components from the skin sample.
- As the reviewer III’s suggestion, the experimental section had been modified and the reference 4 related to ScCO2 had been cited in the revised manuscript. The methods such as FTIR and SEM had been added in the M&M section of manuscript.
Reference 4 is a Review paper, and not clearly describing the process of ScCO2. I recommend adding a couple of papers that really used ScCO2 for the preparation of dECMs.
- As the reviewer III’s suggestion, the macroscopic picture had been added in the Figure 3 of revised manuscript.The SEM were important for the observation of micro-scaffold in this study.
Figure 3 now provides a nice overview over the general porosity of the scaffold.
- As the reviewer III’s suggestion, the descriptions had been modified to avoid confusion such as "The averaged diameter of microscafford was found in a range of 8~25 μm"). The study would like to provide a composite scaffold membrane containing micro-scaffolds instead of macroscoipic scaffolds. Therefore, the designed dECM had been prepared and introduced in the composite scaffold membrane to provide a microstructure of micro-scaffolds. In this study, the micro-scaffolds could be prepared efficiently by using supercritical fluid treatments instead of macroscoipic bioprinting scaffolds. As we know, the introductuion of micro- scaffolds in the bioprinted composted materials are important for regenerative applications. The composite scaffold membrane would be prepared by lyophilisation as experimental statements. The thickness of boundaries of micro-scaffolds within a dried alginate-based composite foam membrane could be obtained. The thickness of boundaries of micro-scaffolds is quite different from those of dried macroscoipic alginate scaffold (hydrated state) as reviewer III’s comments. The statements were not clear as reviewer III’s comments. The statements about “macroscoipic alginate scaffold” and “micro-scaffold” had been modified and described clearly to avoid confusion.
In my comment, I wanted to say, that not the average diameter of the microscaffold was determined, but maybe the average diameter of the fibrous structure. Please clearly describe and improve this section what is seen on the images. Please also give a size range with standard deviation.
- As the reviewer III’s suggestion, the descriptions about thermal analysis had been modified and stated clearly to avoid confusion. It is important to build up a simple method for evaluation of designed bioprintable materials and bioprinted materials derived from bioinks. In this study, the thermal analysis was employed and clear and easy identification was obtained through TGA results(DTG curves). The results could easily identify the molecular interactions between alginate, calcium and the dECM in the bioprinted membrane, which affect the stability and physicochemical properties( such as zeta-potential, viscosity and mechanical stability) of the bioprinted macroscopic scaffold and bioinks.
From only TGA analysis, it is not clear, if the scaffolds will be stable (also by bioprinting). The author does not show if, for example, the scaffolds that are not treated with CaCl2 will be stable, when water is added. This would be important to find out, if these scaffolds could be used to perform cell viability/cell attachment/ cell proliferation studies on them, where the scaffolds would be immersed in cell media for a long time.
Other comments:
-
- Title: the title wants to add too many details, but at the end, it sounds like that ScCO2 was applied on the whole composite scaffold (and not just for the preparation of the dECM). I have the feeling, the author does not really know, what research question he actually wants to answer.
- Abstract L12: please define series.
- Might there be other polymeric additives, other than alginate, that were used in combination with dECM? Please provide some information in the introduction section.
- The differences in the FTIR spectra are not fully understandable, since only amide bonds are marked in the spectra. Please provide the other important peaks for carboxylic acids for example.
- Figure 4: Please provide lower magnifications of these samples, since a high magnification does not make sense in this case to show the differences between the amount of dECM. In my opinion, an overview image would provide more information.
- Figure 5: A and B look to me exactly the same. Why don't you see any differences when 1% CaCl2 was added compared to the sample without calcium? Furthermore, the figure caption is misleading, since no composite membranes were used in these TGA figures.
- In general, the TGA curves do not really provide a clear overview what is going on. There are too many TGA curves in different Figures. It is really difficult to compare for example ALG/dECM4H with ALG/dECM4L with ALG/dECM4N. From my point of view, making a Figure showing the DTG of the three would provide a much better picture of the differences.
- Also the discussion of the TGA data together with the proposed models is very difficult to follow. The author should reformulate the whole section by focusing on the important aspects only. Also, there are a lot of repetitions.
- L336: "a series of" what was obtained? It is not clear what was varied. Also, the whole sentence does not make sense. As it is written now, ScCO2 was applied on the composite scaffold?
- L341: why is high thermal stability important? Please describe.
- Why is "bioprinting" not mentioned in the conclusion section anymore?
Author Response
- Reviewer III’s Comment: I thank the author for addressing my comments. However, I am still skeptical about the quality of this study. The introduction is really messy, and the author is not coming to the main point of the study. In almost every paragraph in the introduction, the aim of the study is described. This should be done in the last paraghraph, clearly describing the aim of the study. At the moment, there are too many repetitions and it is difficult to read.
As the reviewer III’s suggestions, the introduction had been modified. The repetitions had been deleted. The objectives of the present manuscript had been combined in the last paragraph.
- Reviewer III’s Comment: Please also see my answers to the author's statements followed by additional comments:
Still, the title contains "bioprintable", but it has not been shown in the paper, that the system is bioprintable. I would at least suggest to include something like "for bioprinting applications" or similar. If the presented system of alginate and dECM should be bioprintable, the study should contain some information on viscosity, at least. I know, that the viscosity of alginate can be tuned by adjusting the concentration or by the addition of calcium, but at the present status, it seems to be more like a hydrogel. But in the presented manuscript, only films of alginate and dECM is presented.
As the reviewer III’s suggestion, the title had been modified in the revised manuscript. The words such as “bioprintable”, “bioprinted” and “bioprinting” have been deleted to avoid confusion.
- Reviewer III’s Comment: Are the composite membranes bioprinted? If so, please include the methods, e.g. which bioprinter has been used? In the methods section, the composite solutions were casted in a mold an freeze dried. I would not call this bioprinting.
As the reviewer III’s suggestion, the words about "bioprinted" and “bioprinting” had been deleted in the revised manuscript to avoid confusion.
- Reviewer III’s Comment: I agree with the author that the use of ScCO2 seems to be a good method to remove cellular components from the skin sample.
As the reviewer III’s suggestion, the description of ScCO2 was corrected only for preparation of dECM in the revised manuscript to avoid confusion.
- Reviewer III’s Comment: Figure 3 now provides a nice overview over the general porosity of the scaffold.
Thank you for reviewer III’s suggestion about Figure 3. Furthermore, the overview over the porosity of the scaffold had also been provided in Figure 4 of revised manuscript.
- Reviewer III’s Comment: In my comment, I wanted to say, that not the average diameter of the microscaffold was determined, but maybe the average diameter of the fibrous structure. Please clearly describe and improve this section what is seen on the images. Please also give a size range with standard deviation.
As the reviewer III’s suggestion, the “the diameter of the microscaffold” had been corrected to be” the diameter of the fibrous structure ” in the revised manuscript. The “average diameter with standard deviation” was difficult to be reasonably obtained. The “average diameter with standard deviation” had been corrected to be “diameter in the range of…”.
- Reviewer III’s Comment: From only TGA analysis, it is not clear, if the scaffolds will be stable (also by bioprinting). The author does not show if, for example, the scaffolds that are not treated with CaCl2 will be stable, when water is added. This would be important to find out, if these scaffolds could be used to perform cell viability/cell attachment/ cell proliferation studies on them, where the scaffolds would be immersed in cell media for a long time.
As the reviewer III’s suggestion, the concept of stable bioprinted membrane had been deleted because the results only form TGA analysis. The statement of “stability of bioprinted membrane” was corrected to “thermal stability of the designed composite membrane”. TGA results could only support the thermal stability of the designed composite membrane as the reviewer III’s suggestion.
8.Reviewer III’s Comment: Other comments:
(1)Reviewer III’s Comment: Title: the title wants to add too many details, but at the end, it sounds like that ScCO2 was applied on the whole composite scaffold (and not just for the preparation of the dECM). I have the feeling, the author does not really know, what research question he actually wants to answer.
As the reviewer III’s suggestion, the title has been modified and described clearly to avoid confusion. The “Supercritical Fluid Treatments” was only connected to “Decellularized Extracellular Matrix” in the modified Title. The “Decellularized Extracellular Matrix/Alginate Bioprinted Composite Membranes via Supercritical Fluid Treatments” had been corrected to be ” Alginate-based Composite Scaffold Membranes with Decellularized Extracellular Matrix via Supercritical Fluid Treatments” in the revised manuscript. Also, the words such as “bioprintable”, “bioprinted” and “bioprinting” have been deleted to avoid confusion.
(2)Reviewer III’s Comment: Abstract L12: please define series.
In this study, the word “Series” (Abstract L12) would like to be employed for alginate-based composite scaffold membranes with various ratios of “Decellularized Extracellular Matrix” such as ALG/dECM1H, ALG/dECM2H, ALG/dECM3H, and ALG/dECM4H in the original manuscript. As the reviewer III’s comments, the word “Series” had been deleted in the revised manuscript. The abstract had been modified to describe clearly about “alginate-based composite scaffold membranes with various ratios of decellularized extracellular matrix” in the revised manuscript.
(3)Reviewer III’s Comment: Might there be other polymeric additives, other than alginate, that were used in combination with dECM? Please provide some information in the introduction section.
As the reviewer III’s suggestion, the information with references about “other polymeric additives were used in combination with dECM” had been added in the introduction section.
(4)Reviewer III’s Comment: The differences in the FTIR spectra are not fully understandable, since only amide bonds are marked in the spectra. Please provide the other important peaks for carboxylic acids for example.
As the reviewer III’s suggestion, the other important peaks such as carboxylic acids had been signed in the FTIR spectra. Some reference lines had been added for observation in the FTIR spectra in the revised manuscript.
(5)Reviewer III’s Comment:Figure 4: Please provide lower magnifications of these samples, since a high magnification does not make sense in this case to show the differences between the amount of dECM. In my opinion, an overview image would provide more information.
As the reviewer III’s suggestion, the lower magnifications of these samples had been provided to observe overview images in the revised manuscript.
(6)Reviewer III’s Comment: Figure 5: A and B look to me exactly the same. Why don't you see any differences when 1% CaCl2 was added compared to the sample without calcium? Furthermore, the figure caption is misleading, since no composite membranes were used in these TGA figures.
As the reviewer III’s suggestion, Figures 5(A) and 5(B) had been checked and corrected in the revised manuscript. Furthermore, the figure caption of Figures 5 “Thermogravimetric analysis of the composite membranes with/without dECM“ had been corrected to be ”Thermogravimetric analysis of the alginate(ALG) and decellularized extracellular matrix (dECM) samples” in the revised manuscript.
(7)Reviewer III’s Comment: In general, the TGA curves do not really provide a clear overview what is going on. There are too many TGA curves in different Figures. It is really difficult to compare for example ALG/dECM4H with ALG/dECM4L with ALG/dECM4N. From my point of view, making a Figure showing the DTG of the three would provide a much better picture of the differences.
As the reviewer III’s suggestion, a designed Figure about DTG of the three samples (ALG/dECM4H, ALG/dECM4L, ALG/dECM4N) had been prepared to provide a much better picture of the differences in the revised manuscript.
(8)Reviewer III’s Comment: Also the discussion of the TGA data together with the proposed models is very difficult to follow. The author should reformulate the whole section by focusing on the important aspects only. Also, there are a lot of repetitions.
As the reviewer III’s suggestion, the proposed models(Figures 9 and 10 in the original manuscript ) had been deleted and the section had been modified by focusing on the important aspects only in the revised manuscript. Some repetitions had been deleted.
(9)Reviewer III’s Comment: L336: "a series of" what was obtained? It is not clear what was varied. Also, the whole sentence does not make sense. As it is written now, ScCO2 was applied on the composite scaffold?
As the reviewer III’s suggestion, the words of "a series of" were deleted. The statement had been corrected to describe “ The ScCO2 was applied on preparation of dECM.”.
(10)Reviewer III’s Comment: L341: why is high thermal stability? Please describe.
As the reviewer III’s suggestion, the important reason for thermal stability was added and described in the revised manuscript which indicates that the membranes will be very stable under body conditions.
(11)Reviewer III’s Comment: Why is "bioprinting" not mentioned in the conclusion section anymore?
As the reviewer III’s suggestion, the statement of "bioprintable" was deleted to avoid confusion and the statement of "for bioprinting applications" was added for the objectives of the study in the conclusion section.

Round 3
Reviewer 3 Report
I thank the author for improving the manuscript. However, it is still very tiring to read, especially the last section about the thermal stability. It is still too long, and I would say it should be summarized within 1 page maximum. One possibility could be to put several TGA graphs into one graph. That would provide a better and faster overview on the differences between the samples. At the moment, there are 22 TGA graphs in the manuscript, this should be summarized much better.
Also, the method ScCO2 is not described in the introduction for the preparation of dECM scaffolds.
The aim of the study is still not clear. The abstract mentions the extraction process to be important, however, in the results, this method was just used to prepare the fiber scaffolds. The author should provide a clear aim of the study. In my opinion, the ScCO2 method is not relevant to study the interaction between the scaffold and alginate.
Author Response
Dear Sir,
Thank the reviewer for professional and useful suggestions.
1.As the reviewer’s suggestion, the last section about the thermal stability had been corrected and summarized within 1 page. The results of 22 TGA graphs had been listed in new Table 2 to provide a better and faster overview on the differences between the samples.
2.As the reviewer’s suggestion, the method ScCO2 had been described in revised“2.2. Preparation of a decellularized extracellular matrix scaffold." The original statements of “2.2. Treatments with supercritical carbon dioxide before preparation of a decellularized extracellular matrix”and “Preparation of a decellularized extracellular matrix”had been combined to a new statement of “2.2. Preparation of a decellularized extracellular matrix scaffold." in the revised manuscript .
3.As the reviewer’s suggestion, the statements of “ the extraction process to be important” in the Abstract and “ScCO2”in Title had been deleted in revised manuscript to avoid confusion and focus on the aim of study because the ScCO2 method is not relevant to study the interaction between the scaffold and alginate.
Thank you for your kind help.
Best regards,
Joseph. C.C.HUANG
Department of Biomedical Engineering, Ming-Chuan University, Taiwan

Round 4
Reviewer 3 Report
I thank the author for the extensive corrections. The table was a good idea, and in this way, the section about the thermal stability could be shortened a lot.
Unfortunately, the data below 100°C is now missing in the TGA graphs (Figure 5). Therefore, I would suggest to start the graphs in Figure 5 from 50 °C or so.
L247: instead of "as shown in Table 2", I would write "and the peak temperatures are summarized in Table 2" or similar.
L248: put an introduction sentence saying that by the addition of all three components (Alginat + dECM + calcium) there are three distinct peaks occurring and assigned as a, b and c (Table 2 and Figure 5).
Author Response
Dear Sir,
I thank the reviewer for several valuable suggestions. As reviewer’ suggestion, some statements and graphs had been modified in the revised manuscript as following:
- The table was a good idea, and in this way, the section about the thermal stability could be shortened a lot. Unfortunately, the data below 100°C is now missing in the TGA graphs (Figure 5). Therefore, I would suggest to start the graphs in Figure 5 from 50 °C or so.
As reviewer’ suggestion, graphs in Figure 5 had been corrected and started from 50 °C in the revised manuscript.
- L247: instead of "as shown in Table 2", I would write "and the peak temperatures are summarized in Table 2" or similar.
As reviewer’ suggestion, "as shown in Table 2" had been changed to “and the peak temperatures of derivative thermogravimetry(DTG) curves were summarized in Table 2.” in the revised manuscript..
- L248: put an introduction sentence saying that by the addition of all three components (Alginat + dECM + calcium) there are three distinct peaks occurring and assigned as a, b and c (Table 2 and Figure 5).
As reviewer’ suggestion, the introduction sentence "By the addition of all three components (Alginate + dECM + Calcium) there were three distinct peaks occurring and assigned as a, b and c (Table 2 and Figure 5).”had been added in the revised manuscript.
Best regards,
Joseph C.C.Huang
